# Dietary Fiber Intake and Gut Microbiota in Human Health

**DOI:** 10.3390/microorganisms10122507

**Published:** 2022-12-18

**Authors:** Jiongxing Fu, Yan Zheng, Ying Gao, Wanghong Xu

**Affiliations:** 1Department of Epidemiology, School of Public Health, Fudan University, 130 Dong An Road, Shanghai 200032, China; 2Yiwu Research Institute, Fudan University, Building V of Zhongfu Square, Yiwu 322000, China; 3State Key Laboratory of Genetic Engineering, Human Phenome Institute and School of Life Sciences, Fudan University, 2005 Songhu Road, Shanghai 200433, China; 4Key Laboratory of Nutrition, Metabolism and Food safety, Shanghai Institute of Nutrition and Health, Chinese Academy of Sciences, 320 Yueyang Road, New Life Science Building, Room A1926, Shanghai 200031, China

**Keywords:** dietary fiber, gut microbiota, diabetes, modulation

## Abstract

Dietary fiber is fermented by the human gut microbiota, producing beneficial microbial metabolites, such as short-chain fatty acids. Over the last few centuries, dietary fiber intake has decreased tremendously, leading to detrimental alternations in the gut microbiota. Such changes in dietary fiber consumption have contributed to the global epidemic of obesity, type 2 diabetes, and other metabolic disorders. The responses of the gut microbiota to the dietary changes are specific to the type, amount, and duration of dietary fiber intake. The intricate interplay between dietary fiber and the gut microbiota may provide clues for optimal intervention strategies for patients with type 2 diabetes and other noncommunicable diseases. In this review, we summarize current evidence regarding dietary fiber intake, gut microbiota modulation, and modification in human health, highlighting the type-specific cutoff thresholds of dietary fiber for gut microbiota and metabolic outcomes.

## 1. Introduction

Dietary fiber is a carbohydrate in plant foods, such as whole grains, vegetables, fruit, and legumes, which have been dominant in human diets for millions of years. From the Paleolithic era, when the hunter-gatherers mainly ate fruit and wild grains, to the agricultural era, when crops began to be cultivated, the ancients consumed more than 100 g of various digestible and indigestible dietary fiber from plants per day [1,2]. During the million years, the human gut microbiota has provided vital nutritional services through digesting lactose and cellulose, degrading toxins, and biosynthesizing vitamins, signal molecules, and other essential substances [3].

In the industrialization age, however, people consumed much less fiber from diets, posing a big challenge for humans to adapt to the profoundly altered dietary pattern and environments. Relative to the highly conserved human genomes, the flexibility of gut microbes enabled their rapid responses to the changed dietary behaviors and guaranteed the establishment of a new symbiosis with humans [3]. Although the alterations in the gut microbiome may facilitate human adaptation to the changing environment from the perspective of evolution, the new human–microbiome symbiosis, much different from that maintained for millions of years, may elicit profound impacts on human health [4]. Alterations in the human gut microbiome have been implicated in a wide range of complex and chronic conditions, including obesity, diabetes, cancers, and cardiovascular disease [5,6,7,8], and may account for the increasing burden of noncommunicable diseases globally.

To better understand the intricate interplay of dietary fiber intake with gut microbiota in human health, we reviewed the bacterial fermentation of dietary fiber in humans and its role in health maintenance. Aspects considered include the type and intake levels of dietary fiber, the fermentation of dietary fiber by the gut microbiota, the impacts of dietary fiber on the gut microbiota, particularly the fiber type-specific effects and respective cutoff thresholds, and the modulations of dietary fiber on gut microbiota and metabolic outcomes of diabetes patients.

## 2. Dietary Fiber and Its Main Types

The proper definition of dietary fiber was highly debated during the last few decades. The controversy focused on oligosaccharides, a type of resistant carbohydrate with 3 to 9 monomeric units (MU). According to the officially published *Guidelines on Nutrition Labelling* (amended in 2009), dietary fiber refers to “carbohydrate polymers with ten or more MUs, which are resistant to hydrolysis by endogenous enzymes and absorption in the small intestine of humans” [9]. However, subsequent investigations observed homogeneous fermentation and physiological activities of indigestible oligosaccharides and polysaccharides that contain similar monosaccharides, providing supportive evidence for oligosaccharides as one type of dietary fiber [10]. Many countries, including China, Japan, the US, Canada, Brazil, and France, and the European Union have accepted the inclusiveness of oligosaccharides as dietary fiber in their official guidelines or standards [11,12].

Dietary fiber can be obtained from diets as edible carbohydrate polymers naturally occurring in the food, or carbohydrate polymers extracted from food raw material by physical, enzymatic or chemical means, or synthetic carbohydrate polymers having a physiological effect of benefit to health [9]. As shown in Figure 1, dietary fiber can be classified into three types based on the physiological properties of their MU polymerization: 1) nonstarch polysaccharides (NSPs) (MU ≥ 10); 2) resistant starches (RS) (MU ≥ 10), and 3) resistant/nondigestible oligosaccharides (ROS) (MU: 3–9) [13,14]. The NSPs mainly include cellulose, hemicellulose, pectins, inulin, and various hydrocolloids [15,16]. Inulin is a fructan containing 2–60 fructose units. When MU <10, inulin is also recognized as fructo-oligosaccharides (FOS) [17], well-documented prebiotics [14]. RS can be further classified into RS 1 to RS 5, which can be derived from milled grains and seeds (RS 1), raw potatoes, maize and green bananas (RS 2), cooked and cooled potatoes and cornflakes (RS 3), bakery products (RS 4) and fried rice chips (RS 5) [18]. ROS consist of 3–9 MUs, many of which were named after polymerized monosaccharides, such as galacto-oligosaccharides (GOS), xylo-oligosaccharides (XOS), and galactosides [19].

The unified recognition of oligosaccharides as dietary fiber greatly broadens the range of healthy indigestible carbohydrates available to humans. This may help to improve the accuracy of measurements in sustainable nutritional surveillance [20] and promote intake of foods rich in fiber naturally or artificially for filling up on fiber for health [10,21].

## 3. Average Levels and Recommended Amounts of Dietary Fiber Intake

Table 1 summarizes the updated average levels and recommended amounts of dietary fiber intake worldwide. Generally, the global average levels range from 15 to 26 g/day, lower than the recommended 20 to 35 g/day in most countries. As shown in Table 1, the highest levels of dietary fiber intake were observed in Northern Europe (e.g., Denmark [22], Norway [23]) Central Europe (e.g., Germany [24]), and Australia [25]), probably due to widespread consumption of whole grain rye, oat, and wheat in these countries [26].

Discrepancies in body size and tolerance to high-fiber diets across populations may also account for differences in average levels of intake and recommendations. For example, in Japan, the average dietary fiber intake is 18.0 g/day in women and 19.9 g/day in men, very close to the domestic recommended intake (18 g/day for women and 21 g/day for men) [29], but much lower than the average levels in Western populations. This may be explained by the relatively smaller bodies of the Japanese [44,45] and their habits of eating rice (refined grain) and seafood [46]. Many countries have adopted energy-adjusted levels of dietary fiber as the recommended amounts. As presented in Table 1, the daily recommended intake of dietary fiber is 14 g/kcal in the US [32], 14.6 g/kcal in Germany [41], and 10 g/kcal in Japan [30].

## 4. Fermentation of Dietary Fiber by the Gut Microbiota

Dietary fibers escape digestion in the upper gastrointestinal tract and are fermented by bacteria in the colon. The degree of polymerization, particle size, solubility, viscosity, and other dietary fiber features may influence fiber fermentability and the bacteria specificity [47,48]. Fibers with low-degree polymerization can be degraded into small molecules in the gut with fast fermentation [47]; small particles are more likely to be exposed to microbial enzymes; while soluble, viscous fibers, with a high capacity for water retention and stool formation and thus limited exposure to microbes, are resistant to fermentation [47]. The various interactions between chains of monomers and enzymes influence the growth of bacteria, leading to the fiber-specific gut microbiota [48]. Generally, the saccharolytic degradation of fiber in the gut depends on the specific microbes rich in carbohydrate-active enzymes (CAZymes), mainly glycoside hydrolases (GHs) and polysaccharide lyases (PLs) [49], the two enzymes not existing in humans. As profiled in Figure 2, dietary fibers can be shared by some strains synergistically through cross-feeding, i.e., the breakdown products of polysaccharides partially hydrolyzed by the primary degraders can be used as substrates for the secondary degraders [50]. For example, inulin-type fructans (ITF) were found to be extracellularly hydrolyzed by *Bifidobacteria* in the human colon, liberating monosaccharides and/or oligosaccharides accessible to the butyrate producers, the secondary degraders [51,52,53]. During the utilization and metabolism of polysaccharides by bacteria, multiple metabolites were generated, including gases (e.g., H_2_, CH_4_, CO_2_), lactate, succinate, and short-chain fatty acids (SCFAs) [54,55].

The most abundant SCFAs are acetate, propionate, and butyrate. SCFAs can be used by gut mucosal cells as energy sources, with butyrate the preferred energy substrate for colonocytes [56,57]. The absorbed SCFAs, on the other hand, are transferred to the circulation via the hepatic portal vein to act as signaling molecules [58]. By binding to free fatty acid receptors (FFARs) or G protein-coupled receptors (GPCRs: GPR41/FFAR3, GPR43/FFAR2, GPR109A) of host cells [59], SCFAs can activate the complicated downstream molecular pathways in liver, brain, lung, pancreas, bones, adipose tissue, and other organs [55,60,61,62,63,64]. SCFAs play crucial regulatory roles in host metabolic homeostasis, immunological processes, maintenance of intestinal barriers, neurobiology, skeletal functions, and suppression of inflammation and carcinogenesis [55], and have been proved to be beneficial for human health. SCFAs were believed to be the crucial molecules that alleviated diabetes via greater postprandial glucagon-like peptide 1 (GLP-1) and fasting peptide YY (PYY) [65].

## 5. Impacts of Dietary Fiber on Gut Microbial Community Structure and Diversity

The fermentation of fiber in the colon is driven by the gut microbiota, and in return, the nutritional substrate of intestinal flora can modulate the structure and diversity of the microbiome. Results are consistent regarding the relationship between dietary fiber intake and the beta diversity of gut microbiota. People on fiber-rich diets (rural/unindustrialized diet, Mediterranean diet, or vegetarian diet) were consistently found to harbor a dramatically different microbial community structure from their counterparts living in developed areas [66,67,68]. Recently, a Chinese adult cohort study revealed a significant correlation between the intake of whole grains and vegetables in the habitual diet and the changes in beta diversity of the gut microbiome [69]. Similar shifts in bacterial composition were consistently observed in most intervention trials administering high-fiber diets [70,71,72,73,74,75,76].

As a diverse ecosystem with microbes functionally compensating for each other is more robust against environmental influences [77], higher microbial richness and evenness (i.e., alpha diversity) may represent a healthy community [78]. However, results are inconsistent on the association between dietary fiber and alpha diversity of gut microbiota. In a comparative study, adults from unindustrialized regions in Papua New Guinea accustomed to a plant-based diet were found to have a higher alpha diversity compared with American adults on Western diets [79]. While agrarian diets high in fruit/legume fiber were beneficial for microbial richness [80], the rapidly decreased dietary fiber in Western diets was believed to cause the loss of intestinal biodiversity [81]. However, among two independent populations from Washington DC and New York, Lin et al. [82] did not observe significant associations of total fiber intake with the Shannon index or evenness index. Long-term diet quality (with more fruit and vegetables) in Chinese adults was positively associated with microbiome alpha diversity [83], but frequent habitual intake of whole grains and vegetables was not found to increase the Shannon index in another Chinese population [69].

Results derived from trials were less consistent. In a trial in healthy Swedish volunteers, no significant intervention effect was observed for a 3-month vegetarian diet on the Shannon index compared with a normally omnivorous diet [84]. In randomized trials administering interventions with specific dietary fibers, the alpha diversity was observed to decrease in some target healthy populations [70,73,74,85,86,87,88,89,90], but was found to remain stable in some others [75,91,92,93]. In our previous study, we also observed reduced alpha diversity among diabetes patients taking a higher level of dietary fiber [94]. It seems that high-fiber diets may lead to the enrichment of specific fiber-digested strains: most are beneficial SCFA producers, which may inhibit the residence or growth of detrimental species and thereby demonstrate a temporary loss of alpha diversity [77].

## 6. Influences of Dietary Fiber on Different Gut Microbes

Dietary fiber intake improved the richness of SCFA producers, but demonstrated type-specific roles in microbial proliferation [72,74,85,88,91,95]. Interventions using inulin [72,87,96,97], guar gum [98], resistant starch [88,91,99], GOS [75,76,95,100], FOS [76], or arabinoxylan oligosaccharide (a kind of FOS) [85,101,102] consistently resulted in an increased abundance of *Bifidobacterium,* while intake of a specific fiber type led to the promotion of *Faecalibacterium*, *Ruminococcus* (particularly for RS), *Lactobacillus* (particularly for fibers containing galactose or fructose units), *Akkermania*, or *Roseburia* (Figure 1). Most microbial changes could be detected after 1- to 2-week interventions, but were found to remain stable throughout the whole period of interventions [75,96,98,101].

Theoretically, the increase in SCFA producers by fiber fermentation in the human colon should be followed by an increase in fecal SCFA concentrations [88,91,100,102]. However, numerous studies demonstrated the opposite result [70,72,75,76,85,86,88,89,90,96,97,103]. The increased fecal bulk due to high-fiber intake may dilute the concentration of SCFAs [104].

### 6.1. Bifidobacterium

The bifidogenic effect of dietary fiber determines its ability to increase various species belonging to the *Bifidobacterium* genus. Healey et al. [72] found that a supplement of inulin increased the relative abundance of *Bifidobacterium* from 6.69% to 15.07%. Kiewiet et al. [96] found that inulin mainly increased the abundance of *Bifidobacterium adolescentis* and raised *Bifidobacterium angulatum* and *Bifidobacterium ruminantium* to a detectable level in the treatment group. The consumption of inulin-rich food was also found to increase the *Bifidobacterium longum* level by threefold [87]. In addition, participants consuming 2-week partially hydrolyzed guar gum had an increased *Bifidobacterium* abundance to 12% from 8% at baseline, while those taking a placebo did not [98]. RS from potato led to a 6.5-fold elevated level of *Bifidobacterium faecale*/*adolescentis*/*stercoris* sequences [91].

GOS was reported to remarkably increase the relative abundance of *Bifidobacterium* from 7.0% to 34.8% [75]. Another study also observed a GOS-induced elevated bifidobacteria count from 7.2 to 7.8 (log 10 CFU/g). Adults treated with GOS and FOS had more abundant *Bifidobacterium*, with linear discriminant analysis effect sizes (LDA effect size) of around 4.0 compared to the baseline level [76]. Müller et al. [85] found that AXOS significantly increased two OTUs of *Bifidobacterium*, the leading drivers of the microbial deviations between pre- and postintervention.

### 6.2. Faecalibacterium

Many long-chain types of fiber demonstrated effects to enhance the abundance of *Faecalibacterium*. Inulin was found to increase the relative abundance of *Faecalibacterium* from 0.41% to 0.61% [72]. Partially hydrolyzed guar gum persistently caused an increment of *Faecalibacterium* during intervention and washout periods [74]. The consumption of whole-grain wheat rich in RS and total fiber induced more than a doubling of *Faecalibacterium* spp. [88]. Similar results were found by Hughes et al. [90]: an RS2-enriched wheat intervention was related to a significant elevation in *Faecalibacterium* compared to baseline and the control group.

### 6.3. Ruminococcus

The specific effect of RS2 on the proliferation of *Ruminococcus* was consistently demonstrated in previous studies. Baxter et al. [91] found that RS2 derived from native maize led to a 2.5-fold increase in the relative abundance of *Ruminococcus bromii*, a specific taxon that performed as a primary degrader of RS. Martínez et al. [99] noticed that subjects consuming maize RS2 had a significantly higher proportion of *Ruminococcus bromii* (average 4.1%) than at baseline (average 1.0%) and those taking placebo (average 2.6%) or chemically modified RS4 (average 1.2%). RS2-enriched wheat/whole-grain wheat was found to induce increases in the *Ruminococcus* genus and *Ruminococcus bromii* [88,90].

A similar *Ruminococcus*-induced effect was observed for other types of fiber. For examples, arabinogalactan supplementation resulted in a nearly eightfold increase in an uncultured *Ruminococcus* spp. [86]. Both Yasukawa et al. [98] and Reider et al. [74] observed an association of partially hydrolyzed guar gum with a bloom in *Ruminococcus*. Regarding inulin, a negative correlation was observed between the consumption and the richness of *Ruminococcus bromii* [91]. A randomized trial of 34 healthy participants demonstrated a decreased relative abundance of *Ruminococcus* (from 2.11% to 1.15%) by intervention with an ITF prebiotic [72]. Administration of yeast mannan, another type of gum, also contributed to a lower abundance of *Ruminococcus* compared with the control group [103].

### 6.4. Lactobacillus

Fibers containing fructose or galactose units such as inulin, fructo-oligosaccharides and galacto-oligosaccharides could particularly lead to a higher fecal abundance of *Lactobacillus*. An intervention using very-long-chain inulin, a special inulin with an average number of fructose units between 50 and 103, was found to increase lactobacilli levels by 2.42-fold compared to baseline and by 5.88-fold relative to the placebo group [97]. Another inulin-induced increment of *Lactobacillus* was observed in 34 healthy subjects (0.26% to 1.26%), and the effect was more pronounced (from 0.6% to 3.0%) in subjects with initial low habitual dietary fiber intake [72]. Deshipu stachyose granules, a mixture of GOS derived from the dietary roots of Lycopus lucidus turcz, were found to increase the mean number of lactobacilli from 6.8 to 7.5 (log 10 CFU/g) after 14-day treatment [95]. A lactobacilli-promoted effect was also observed for gum arabic [105] and arabinoxylan oligosaccharides (AXOS) [85].

### 6.5. Prevotella

Observational studies consistently identified a specific higher *Prevotella* in unindustrialized or rural populations consuming high-fiber diets, just like Hadza hunter-gatherers in Tanzania [106] and adults from Papua New Guinea [79], than those from industrialized regions or urban areas [66,79,106,107]. Therefore, Wu et al. [108] proposed the concept of gut microbial enterotypes, based on which gut microbiomes can be clustered as *Prevotella* enterotype (corresponding to dietary carbohydrates) and *Bacteroides* enterotype (corresponding to dietary protein and animal fat).

However, many intervention trials did not find that *Prevotella* could be induced by dietary fiber [108,109]. The discrepancy might be due to the ability of *Prevotella* to ferment complicated carbohydrates, not only indigestible fiber but also digestible carbohydrates, digested oligosaccharides, and monosaccharides [108], all of which were rich in the typical starchy plant food in unindustrialized areas [110]. It is the long-term intake of various carbohydrates in habitual diets but not the short-term single-fiber intervention that could construct a *Prevotella*-dominated microbial community [109,111]. The microbial differences could also be explained by the discrepancies in geography, ethnicity, and lifestyles across populations [112].

In summary, low-fiber diets have been suggested to influence the richness of the gut microbiome in healthy individuals, disrupt the symbiotic relationship between the gut microbiota and the intestine, and may increase the risk of diseases. High-fiber diets, on the other hand, have been used to modify the microbiota to achieve improved health outcomes.

## 7. Cutoff Threshold of Dietary Fiber Intake on Gut Microbiota

A dose-dependent bacterial proliferation with levels of fiber was demonstrated in vitro with significant growth, but only with detected metabolic promotion at a low level [113]. The relationship was also observed in population studies. Among 80 adults with an average of 14.1 ± 5.11 g/day intake of dietary fiber, Dominianni et al. [114] observed different compositions of gut microbiota in subjects with more than 11.7 g/day dietary fiber intake and found linear negative associations of fiber intake with *Coprococcus* and *Porphyromonadaceae*. Gaundal et al. [115] observed an inverse linear association of *Alistipes* with dietary fiber among subjects consuming more than 30 g/day dietary fiber and a positive correlation of *Bacteroides stercoris* with a total intake of healthy food components, such as fiber and grain products. In our previous study on diabetes patients, a mediating effect was observed for *Desulfovibrio* in the habitual dietary fiber–A_1c_ associations among patients taking more than 7.2 g/day dietary fiber, but not in those taking less [94].

In intervention trials, administering multiple dosages of dietary fiber facilitate the evaluation of the type-specific dose-dependent microbial effects of dietary fiber and enabled the identification of cutoff thresholds. Summarized in Table 2 are the trials implementing multiple-dosage inulin, gum arabic, RS4, FOS, AXOS, or resistant maltodextrin. These trials demonstrated various altered taxa in gut microbiota induced by dietary fiber, but consistently observed an increased abundance of bifidobacteria, albeit with varied cutoff thresholds of fiber intake.

In the report of Kolida et al. [116], 25 of 30 individuals taking 8 g/day inulin responded positively and only 20 of 30 in the group taking 5 g/day. In Reimer et al.’s study [117], subjects on 7 g/day ITF had higher levels of *Bifidobacterium*, *Cellulomonas*, *Nesterenkonia*, and *Brevibacterium* and lower *Lachnospira* and *Oscillospira*, with a cutoff threshold of 3 g/day for *Bifidobacterium* only. Calame et al. [105] used multiple dosages of gum arabic (0, 5, 10, 20, 40 g/day for 4 weeks) and found significantly increased bifidobacteria, lactobacilli and bacteroides in the volunteer group consuming 10 g/d gum arabic only, but not those on a higher dosage. Deehan et al. [118] observed remarkable microbial modulations by RS4 derived from maize and tapioca, particularly for the dosages of 20 g/day for maize and 35 g/day for tapioca.

The dose-dependent effect of fructo-oligosaccharides (FOS), a prebiotic supplement with bifidogenic capabilities, has been well evaluated. Bouhnik et al. [119] administrated 0 to 20 g/day short-chain FOS (SC-FOS) in healthy volunteers, and observed a dose-dependent increase in fecal bifidobacteria with a cutoff value of 5 g/day. Considering the tolerance of humans, the optimal dose of SC-FOS is 10 g/d for significantly elevated fecal bifidobacteria in healthy volunteers consuming their usual diet. In a subsequent trial, Bouhnik et al. [120] observed a linear correlation of bifidobacteria, with SC-FOS ranging from 0 to 10 g/day. Tandon et al. [121] identified multiple beneficial microbes promoted by FOS, including *Bifidobacterium*, *Lactobacillus* (predominated by *Lactobacillus ruminis*, which was newly classified as *Ligilactobacillus ruminis* [127]), *Faecalibacterium*, and *Ruminococcus*, with relatively higher increases at 10 g/day than the other three levels (0, 2.5, 5 g/day). The microbial effects of AXOS were evaluated in healthy men and women by Maki et al. [122] and François et al. [123], respectively. *Bifidobacterium* spp. and postprandial plasma ferulic acid were found to be significantly higher in subjects consuming 4.8 g/d AXOS than those on 2.2 g/d or 0 g/d (control) [122]. In comparison, bifidobacteria and SCFAs (acetic acid and propionic acid) increased after consumption of 8 g/day AXOS (not significant at 2.4 g/day) [123].

The microbiological properties of resistant maltodextrin were incompletely understood. No significant boost in *Bifidobacterium* but a linearly increased proportion of propionic acid in stool samples were observed in a trial administering 0, 7.5, 15 g/day resistant maltodextrin [104]. Lefranc-Millot et al. [124] did not find altered abundance of *Bifidobacterium* spp. or *Lactobacillus* spp., but apparent dose-dependent effects of resistant dextrin on *Bacteroides* (increased at 10 g/d only), *Clostridium perfringens* (decreased at 15 g/d only), and colonic pH (changed at 20 g/d only). Based on the same database, two studies obtained different results. In one study, only 25 g/day resistant maltodextrin consumption resulted in a positive change in bifidobacteria [125], while another analysis observed a slightly increased abundance of *Akkermansia muciniphila* and *Faecalibacterium prausnitzii* among subjects taking 25 g/d resistant maltodextrin, but only among those with low baseline abundance of the bacteria [126]. It seems that the initial abundance of microbes resident in the human colon may determine the effects of fiber interventions [72,101,116]. Cloetens et al. [101] found that—regardless of the dosage used and the duration of intervention—AXOS supplementation succeeded in increasing *Bifidobacterium* in subjects with a low initial abundance, but failed in those without any detected fecal *Bifidobacterium* at baseline, further supporting the influences of the initial abundance of microbes.

## 8. Modulation of Dietary Fiber on Gut Microbiota in Diabetes Patients

Alterations in the human gut microbiota can be implicated in a variety of complex and chronic diseases, such as diabetes, obesity, cancers, and cardiovascular disease [5,6,7,8]. Type 2 diabetes is a metabolic disease that may affect and can be affected by gut microbiota [128], which represents a good example to show how dietary fiber induces modulation of the gut microbiota and improves clinical outcomes.

Table 3 presents selected trials administering fiber interventions among diabetes patients. The first trial focused on the effects of GOS by administering an intervention with low-dose GOS (5.5 g/day for 12 weeks) among 29 men with well-controlled type 2 diabetes [129]. As a result, a slightly elevated abundance of *Bifidobacterium* was observed, but without significant overall changes in gut microbiota or any improvement in glucose metabolisms. In 52 Japanese patients with type 2 diabetes, Gonai et al. [130] observed markedly restored abundance (taxa at the family level only) of *Bifidobacteriaceae* by GOS, but found reduced OTUs, *Lachnospiraceae*, *Ruminococcaceae*, *Peptostreptococcaceae*, *Erysipelotrichaceae*, and *Porphyromonadaceae*. Although no significant improvements in metabolic outcomes were observed in this study, the negative correlations of HbA_1c_ levels with abundance of *Bifidobacteriaceae* and *Peptostreptococcaceae*, fasting plasma glucose (FPG) with *Peptostreptococcaceae,* and triglyceride (TG) with *Ruminococcaceae*, and the positive correlations of aspartate transaminase (AST) and alanine transaminase (ALT) with *Lachnospiraceae* indicate the roles of gut microbiota in metabolic homeostasis among diabetes patients.

In a milestone intervention trial conducted in Chinese diabetes patients, Zhao et al. [65] found that a diet with a high mixture of fibers selectively promoted SCFA-producing strains, particularly fecal *Bifidobacterium* spp., leading to enriched CAZyme-encoding genes for starch and inulin degradation and activated pathways for acetate and butyrate formation, and thus achieved a better improvement in HbA_1c_ levels and postprandial insulin concentrations. More trials were subsequently designed to evaluate the intervention effects of specific types of dietary fiber. In a pilot study of 22 high-risk adults for type 2 diabetes, Mitchell et al. [132] observed significantly increased bifidobacteria and reduced fasting insulin level and homeostatic model of assessment for insulin resistance (HOMA-IR) after a 6-week inulin (10 g/day) intervention, but did not find a significant correlation of changes in bifidobacteria with any metabolic outcomes. In another pilot study, Mateo-Gallego et al. [134] added isomaltulose (16.5 g/day) and resistant maltodextrin (5.28 g/day) into alcohol-free beer for 14 diabetes subjects for 10 weeks, and observed higher *Parabacteroides* and lower *Bacteroides, Odoribacter, Butyricimonas* and *Oscillospira* accompanying decreased body mass index, blood glucose level and HOMA-IR.

In recent years, well-designed RCTs have supported beneficial alternations in gut microbiota induced by specific types of fiber in diabetes patients. A one-year intervention using a soluble viscous fiber product containing 15–20 g/day dietary fiber achieved significantly increased *Collinsella*, *Parabacteroides* and *Roseburia*, but decreased *Faecalibacterium*, *Lactobacillus*, and *Oscillibacte*, and thus improved metabolic outcomes of diabetes patients, including decreased levels of body mass index, waist circumference, HbA_1c_ and LDL [133]. Birkeland et al. [131] evaluated the prebiotic effect of ITF on the fecal microbiota and produced SCFAs in 25 patients with type 2 diabetes. A prominent bifidogenic effect was observed, with the highest positive effect on OTUs of *Bifidobacterium adolescentis*, followed by OTUs of *Bacteroides*. Analogous effects were also observed among prediabetes individuals administered beta glucan, inulin, RS and GOS [135,136].

Evidently, altering the gut microbiota of diabetes patients has great potential to improve glycemic control status, and can be used as an effective intervention for patients [137]. However, alpha diversity of gut microbiota was observed to remain unchanged or decrease in patients placed on a high-fiber diet. *Bifidobacterium* was the only beneficial fiber-enriched microbe consistently induced by fiber interventions. It is possible that—due to prevalent diabetes or related conditions—the gut microbiome may have been altered, leading to serious depletion of beneficial bacteria and microbial dysbiosis that could not be easily promoted by dietary interventions [128,138,139]. As the key roles of SCFAs in metabolic enhancement have been highlighted, more approaches should be developed and validated to improve the abundance of microbial SCFAs producers.

## 9. Conclusions and Future Prospects

Evidence is accumulating on the beneficial effects of dietary fiber intake on human health. The mechanisms involved have been better understood in recent years, indicating the critical role of the gut microbiota in this process via the production of SCFAs and other functional metabolites. The decreasing dietary fiber intake over the centuries has fostered a gut microbiota detrimental to human health, leading to a global epidemic of diabetes, cancers, and other noncommunicable diseases. The responses of the gut microbiota to increased availability of dietary fiber may differ by type, level, and duration of intake, demonstrating dietary fiber type-specific cutoff thresholds. Understanding the intricate interplay between dietary fiber and the gut microbiota may help to develop effective intervention strategies to prevent and control noncommunicable diseases. Further studies are warranted to further elucidate this complex relationship in human health and identify targets for effective interventions. 

## Figures and Tables

**Figure 1 microorganisms-10-02507-f001:**
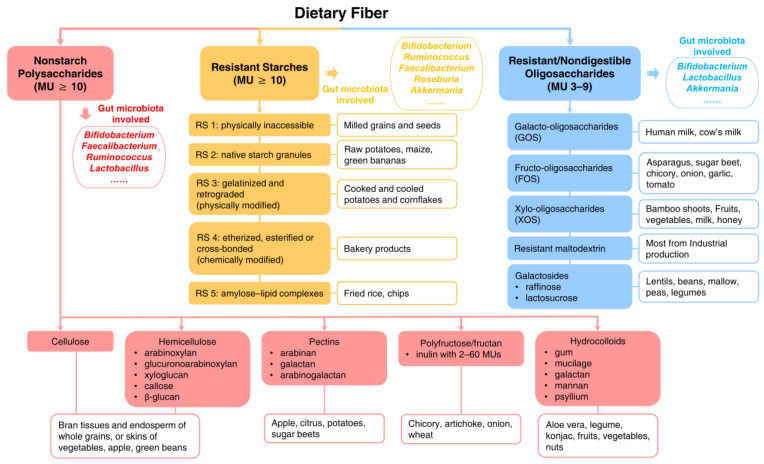
Type of dietary fiber. MU: monomeric unit.

**Figure 2 microorganisms-10-02507-f002:**
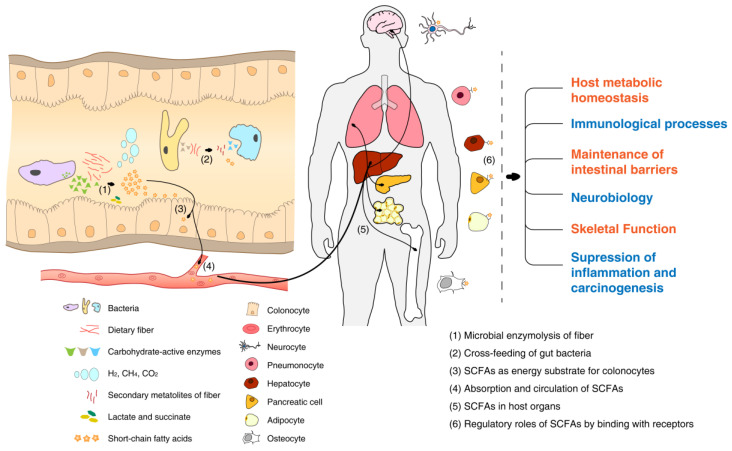
Fermentation of dietary fiber by gut microbiota.

**Table 1 microorganisms-10-02507-t001:** Global intake and recommendations of dietary fiber.

Region	Intake Level (g/day)			Reference	Recommendation (g/day)			Reference
	Men	Women	Overall		Men	Women	Overall	
**Asia**								
China	19.4	17.6	--	China Health and Nutrition Survey (2011) [27]	--	--	25–30	Chinese Dietary Reference Intake (2017) [28]
Japan	19.9	18.0	18.8	National Health and Nutrition Survey in Japan (2019) [29]	21	18	10 g/kcal	Dietary Reference Intake for Japanese (2020 Edition) [30]
**North America**								
USA	18.1	15.2	16.6	What We Eat in America (2017–2020) [31]	38	25	14 g/kcal	Dietary Guidelines for Americans (2020–2025) [32]
Canada	18.4	16.2	--	Canadian Community Health Survey (2015) [33]	38	25	--	Canada’s Dietary Guidelines (2019) [34]
**Oceania**								
Australia	24.8	21.1	--	Australian Health Survey (2011–2012) [25]	38	28	--	Australian Dietary Guidelines (2013) [35]
**Europe**								
European Union					--	--	25	Scientific Opinion on Dietary Reference Values (EFSA-2010) [13]
UK	--	--	19.7	UK National Diet and Nutrition Survey (2016–2019) [36]	--	--	30	Eatwell Guide (2016) [37]
France	21.6	17.7	19.6	National Individual Food Consumption 3 (2014–2015) [38]	--	--	30	Guidelines: development of nutritional references (2016) [39]
Denmark	--	--	22	Dietary habits in Denmark (2011–2013) [22]	30	25	30 g/10MJ	Nordic Nutrition Recommendations (2012) [40]
Norway	26	22	24	Norkost 3 (2010–2011) [23]				
Germany	24.8	23.1	--	National Consumption Research II (2005–2007) [24]	--	--	30 or 14.6 g/kcal	The D-A-CH reference for nutrient intake (2021) [41]
Russia	15	12	--	Russia Longitudinal Monitoring Survey (2018) [42]	--	--	30	Center for hygienic education of the population [43]

**Table 2 microorganisms-10-02507-t002:** Cutoff thresholds of dietary fiber on gut microbiota in intervention trials.

Author (Year) Region	Participants N | F:M ^a^ Mean Age (Years)	RCT Design	Dietary Fiber	Type of Fiber	Administrated Dosage (g/day)	Duration of Intervention	Changes in Gut Microbiota ^b^	Cutoff Threshold (g/day)
Kolida, S. [116] (2007) UK	30 | 15:15 26.5 ± 3.1	double-blind crossover	Inulin	NSP	0 g 5 g 8 g	2-week intervention 1-week washout	↑: Bifidobacteria * Depending on initial abundance	NO
Reimer, R. A. [117] (2020) Canada	50 | 28:22 around 31	double-blind crossover	Inulin-type fructans (ITF)	NSP	0 g or 7 g (n = 25) 0 g or 3 g (n = 25)	4-week intervention 4-week washout	↔: α-diversity (Shannon index) ↔: β-diversity (Bray–Curtis distance) ↑: *Bifidobacterium*, *Cellulomonas*, *Nesterenkonia*, *Brevibacterium* ↓: *Lachnospira*, *Oscillospira* ↔: Fecal SCFAs	7 g for *Bifidobacterium*, *Cellulomonas*, *Nesterenkonia*, *Brevibacterium*, *Lachnospira*, *Oscillospira*
Calame, W. [105] (2008) Netherlands	48 | -- 30.9 ± 12.8	double-blind	Gum arabic	NSP	0 g (placebo-control, n = 8) 5 g (n = 8) 10 g (n = 8) 20 g (n = 8) 40 g (n = 8) 10 g inulin (positive control, n = 8)	4-week intervention	↑: Bifidobacteria, lactobacilli, bacteroides	10 g for bifidobacteria, lactobacilli, bacteroides
Deehan, E. C. [118] (2020) Canada	40 | 20:20 28.4 ± 8.1	double-blind	RS4	RS	Placebo (n = 10) Maize RS4 (n = 10) Potato RS4 (n = 10) Tapioca RS4 (n = 10) All 10 g for 1 week →20 g for 1 week →35 g for 1 week →50 g for 1 week	4-week intervention	In maize group: ↓: α-diversity (Pielou and Shannon index) ↕: β-diversity (Bray-Curtis) ↑: *Bifidobacterium* ↑: *Eubacterium rectale*, *Oscillibacter*, *Anaeromassilibacillus*, *Ruminococcus* ↓: *Agathobaculum butyriciproducens*, *Adlercreutzia equolifaciens* ↑: Fecal butyrate In tapioca group: ↓: α-diversity (Pielou and Shannon index) ↕: β-diversity (Bray-Curtis) ↑: *Bifidobacterium* ↑: *Parabacteroides distasonis*, *Faecalibacterium*, *Eisenbergiella* ↓: *Eubacterium hallii* and *Clostridium viride* ↑: Fecal propionate *No effect of potato RS4 detected	In maize group: 20 g for α-diversity, β-diversity, *Eubacterium rectale*, *Oscillibacter*, *Anaeromassilibacillus*, *Ruminococcus*; 35 g for fecal butyrate; In tapioca group: 35 g for α-diversity, β-diversity, fecal propionate
Bouhnik, Y. [119] (1999) France	40 | 22:18 29.6		Short-chain fructo-oligosaccharides (SC-FOS)	ROS	0 g (n = 8) 2.5 g (n = 8) 5 g (n = 8) 10 g (n = 8) 20 g (n = 8)	7-day intervention	↑: Bifidobacteria	5 g for bifidobacteria
Bouhnik, Y. [120] (2006) France	40 | 22:18 29 ± 1.3		Short-chain fructo-oligosaccharides (SC-FOS)	ROS	0 g (n = 8) 2.5 g (n = 8) 5 g (n = 8) 7.5 g (n = 8) 10 g (n = 8)	7-day intervention	↑: Bifidobacteria (linear correlation) ↑: Total anaerobes	10 g for total anaerobes
Tandon, D. [121] (2019) India	69 | 35:34 around 30	double-blind	Fructo-oligosaccharides (FOS)	ROS	0 g (n = 17) 2.5 g (n = 16) 5 g (n = 18) 10 g (n = 18)	90-day intervention	↑: α-diversity (Chao) ↑: *Bifidobacterium*, *Lactobacillus*, *Faecalibacterium*, *Ruminococcus*, *Sutterella*, *Oscillospira* *Reversal impact of prebiotics postdiscontinuation *Inconsistent results of analyses performed on data at two distinct levels of taxonomy (OTUs or genus)	10 g for α-diversity, *Bifidobacterium*, *Lactobacillus*, *Faecalibacterium*, *Ruminococcus*, *Sutterella*, *Oscillospira*
Maki, K. C. [122] (2012) The US	55 | -- 53.1 ± 12.6	double-blind crossover	Arabinoxylan-oligosaccharide (AXOS)	ROS	0 g 2.2 g 4.8 g	3-week intervention 2-week washout	↑: *Bifidobacterium* spp.	4.8 g for *Bifidobacterium* spp.
François, I. E. [123] (2012) Belgium	57 | 27:30 42 ± 17	double-blind crossover	Arabinoxylan oligosaccharides (AXOS)	ROS	0 g 2.4 g 8 g	3-week intervention 2-week washout	↑: Bifidobacteria ↑: Fecal SCFAs (acetic acid, propionic acid)	8 g for bifidobacteria, fecal SCFAs
Fastinger, N. D. [104] (2008) The US	38 | 19:19 around 27	double-blind	Resistant maltodextrin	ROS	0 g (n = 12) 7.5 g (n = 13) 15 g (n = 13)	3-week intervention	↑: *Bifidobacterium* (nonsignificant) ↑: Proportion of propionic acid (linear)	NO
Lefranc-Millot, C. [124] (2012) France	48 | 24:24 28	double-blind	Resistant dextrin	ROS	0 g(n = 12) 10 g (n = 13) 15 g (n = 12) 20 g (n = 11)	2-week intervention	↔: *Bifidobacterium* spp., *Lactobacillus* spp. ↑: *Bacteroides* ↓: *Clostridium perfringens* ↓: Colonic pH	10 g for *Bacteroides*; 15 g for *Clostridium perfringens*; 20 g for colonic pH
Burns, A. M. [125] (2018) The US	49 | 28:21 26.3 ± 6.8	double-blind cross-over	Resistant maltodextrin	ROS	0 g (n = 16) 15 g (n = 17) 25 g (n = 16)	3-week intervention 2-week washout	↑: Bifidobacteria	25 g for bifidobacteria
Mai, V. [126] (2022) The US	49 | 28:21 26.3 ± 6.8	double-blind cross-over	Resistant maltodextrin	ROS	0 g (n = 16) 15 g (n = 17) 25 g (n = 16)	3-week intervention 2-week washout	↑: *Fusicatenibacter saccharivorans* ↑: *Akkermansia muciniphila*, *Faecalibacterium prausnitzii* (in individuals with low baseline counts)	25 g for *Akkermansia muciniphila*, *Faecalibacterium prausnitzii*

^a^: Ratio of female subjects to male subjects; -- data not available. ^b^: ↑ increased; ↓ decreased; ↔ unchanged; ↕ changed β-diversity.

**Table 3 microorganisms-10-02507-t003:** Modulation of dietary fiber on gut microbiota and metabolic outcomes in patients with type 2 diabetes.

Author Year Region	Subjects	F:M ^a^ Mean Age (years)	RCT Design	Fiber Type or Sources	Amount of Fiber (g/day)	Duration of Intervention	Change in Gut Microbiota ^b^	Metabolic Effects ^b^	Microbiome and metabolic Indicators
Pedersen, C. [129] (2016) UK	29 well-controlled men	All men around 57	Double-blind	galacto-oligosaccharide (GOS)	5.5 (n = 14) placebo (n = 15)	12 weeks	↑: α-diversity (Shannon and Simpson indices) ↑: *Bifidobacterium* (close to significance)	No significant effect on glucose, insulin, or C-peptide fasting concentrations	*Bifidobacterium* positively correlated with total AUC of glucose and IL-6
Gonai, M. [130] (2017) Japan	52 patients	-- 50 ± 10	Double-blind	galacto-oligosaccharide (GOS)	10 (n = 27) placebo (n = 25)	4 weeks	↓: α-diversity (Observed OTUs) ↑: *Bifidobacteriaceae* ↓: *Lachnospiraceae*, *Ruminococcaceae*, *Peptostreptococcaceae*, *Erysipelotrichaceae*, *Porphyromonadaceae*	No clinical parameters changed significantly	Negative correlations of A_1c_ with *Bifidobacteriaceae* and *Peptostreptococcaceae;* FPG with *Peptostreptococcaceae*; TG with *Uminococcaceae*; positive correlations of AST and ALT with *Lachnospiraceae*
Zhao, L. [65] (2018) China	43 patients	-- 35–70	Open-label	fiber in diet	high-fiber diet (n = 27) usual care (n = 16)	84 days	↕: β-diversity (Bray-Curtis) ↑: *Bifidobacterium spp.* and other SCFA producers ↑: CAZyme-encoding genes for starch and inulin degradation ↑: *fhs* for acetate and *but* for butyrate formation pathway ↑: acetate and butyrate	↓: A_1c_ ↑: Postprandial insulin	Higher acetate and butyrate coincided with a significantly greater AUC of postprandial glucagon-like peptide-1 and a higher level of fasting peptide YY, which partly improve A_1c_ level
Birkeland, E. [131] (2020) Norway	25 patients	10:15 63.1	Double-blinded, crossover	inulin-type fructans	16 placebo	6-week intervention 4-week washout	↔: α-diversity (observed OTUs) ↕: β-diversity ↑: *Bifidobacterium adolescentis, Bacteroides ovatus, Faecalibacterium prausnitzii* ↓: *Ruminococcus* ↑: SCFAs (acetic acid and propionic acid)		*Bifidobacterium adolescentis* negatively related to fecal butyric acid
Mitchell, C. M. [132] (2021) US	22 adults at risk for T2D	14:8 54.4 ± 8.3	Double-blind	Inulin	10 (n = 13) placebo (n = 9)	6 weeks	↑: Bifidobacteria	↓: Fasting insulin, HOMA-IR	No significant correlation between changes in bifidobacteria and any outcome variables.
Reimer, R. A. [133] (2021) Canada	290 adult patients with overweight/obesity	198:92 around 55	Double-blind	soluble viscous fiber	15–20 (n = 147) isocaloric placebo (n = 143)	52 weeks	↑: *Collinsella*, *Parabacteroides, Roseburia* ↓: *Faecalibacterium*, *Lactobacillus, Oscillibacter*	↓: A_1c_, BMI, WC, LDL	
Mateo-Gallego, R. [134] (2021) Spain	14 patients with overweight or obesity	-- 56.1 ± 6.27	Double-blinded, crossover	isomaltulose + resistant dextrin	16.5 + 5.28 placebo	10-week intervention 6–8 weeks’ washout	↔: α-diversity (Shannon, Pielou, Observed features) ↔: β-diversity (weighted Unifrac) ↑: *Parabacteroides* ↓: *Bacteroides*, *Odoribacter*, *Butyricimonas, Oscillospira*	↓: BMI, Blood glucose, HOMA-IR	

^a^: Ratio of female subjects to male subjects; -- data not available. ^b^: ↑ increased; ↓ decreased; ↔ unchanged; ↕ changed β-diversity.

## Data Availability

Not applicable.

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
