# Peer review of "Dietary Fiber Intake and Gut Microbiota in Human Health"

_microorganisms, 2022, doi:10.3390/microorganisms10122507_

Round 1

Reviewer 1 Report

The proposed manuscript for review article explores the relationship between dietary fibre, gut microorganisms and human health. The topic  is well defined in the title and abstract section. The introduction includes the purpose of the review. The manuscript as a whole is well written and informative. it summarizes and discusses the available data appropriately. The language is clear and concise.

I recommend that the manuscript is accepted for publication without revision.

Author Response

Point 1: The proposed manuscript for review article explores the relationship between dietary fibre, gut microorganisms and human health. The topic is well defined in the title and abstract section. The introduction includes the purpose of the review. The manuscript as a whole is well written and informative. it summarizes and discusses the available data appropriately. The language is clear and concise.

I recommend that the manuscript is accepted for publication without revision.

Response 1: We greatly appreciate the reviewer’s favorable statement.

Reviewer 2 Report

Dear authors

The idea of this review is very important due to the importance of dietary fiber to human health and its role in the enhancement of gut microbiota growth, which play valuable role in protection of human from   chronic diseasease.

Author Response

Point 1: The idea of this review is very important due to the importance of dietary fiber to human health and its role in the enhancement of gut microbiota growth, which play valuable role in protection of human from chronic disease.

Response 1: Thank the reviewer for his / her kind recognition.

Reviewer 3 Report

In this study, the authors summarized current evidence regarding dietary fiber intake, gut microbiota modulation, and modification in human health, highlighting the type-specific cut-off thresholds of dietary fiber for gut microbiota and metabolic outcomes. This topic is interesting for experts in the corresponding fields. However, there are some problems not being explained clearly in the MS. The authors only listed the results of others, but did not summarize the results and put forward his own opinions, such as the reason and mechanism.

Specific comments:

1. In the part of “4. Fermentation of dietary fiber by the gut microbiota ”, the author only referred that “The degree of polymerization, particle size, solubility, viscosity, and other dietary fiber features may influence fiber fermentability and the bacteria specificity”. It is necessary to explain how the degree of polymerization, particle size, solubility, viscosity, and other dietary fiber features may influence fiber fermentability and the bacteria specificity. Moreover, the author referred that dietary fiber can be classified into three types based on the physiological properties of their MU polymerization: 1) nonstarch polysaccharides (NSP) (MU10); 2) resistant starches (RS) (MU10), and 3) resistant/nondigestible oligosaccharides (ROS) (MU: 3-9) . When different types of dietary fiber are fermented by intestinal flora, what are the differences in intestinal flora that ferment the dietary fiber and the produced metabolites?

2.In the part of “5. Impacts of dietary fiber on gut microbial composition and diversity”, it is necessary to introduce the specific changes of gut microbial composition.

3.In the part of “6. Influences of dietary fiber on different gut microbes ”, the authors only introduced the changes of gut microbes. How about the changes of metabolites produced by gut microbes? What is the significance of these changes and what do these changes mean?

4.In the part of Table 2, the result description in table 2 is too simple.

5.In the part of “7. Cut-off threshold of dietary fiber intake on gut microbiota ”, only the cut-off threshold of Bifidobacterium was introduced. How about other gut microbiota?

6.In the part of “Modulation of dietary fiber on gut microbiota in diabetes patients”, Since alterations in the human gut microbiome have been implicated in a wide range of complex and chronic conditions, including obesity, diabetes, cancers, and cardiovascular disease, why did the authors only introduce the modulation of dietary fiber on gut microbiota in diabetes patients? How does dietary fiber modulate gut microbiota in patients with other diseases, such as obesity, cancers, and cardiovascular disease?

Author Response

Comments: In this study, the authors summarized current evidence regarding dietary fiber intake, gut microbiota modulation, and modification in human health, highlighting the type-specific cut-off thresholds of dietary fiber for gut microbiota and metabolic outcomes. This topic is interesting for experts in the corresponding fields. However, there are some problems not being explained clearly in the MS. The authors only listed the results of others, but did not summarize the results and put forward his own opinions, such as the reason and mechanism.

Point 1.1: In the part of “4. Fermentation of dietary fiber by the gut microbiota”, the author only referred that “The degree of polymerization, particle size, solubility, viscosity, and other dietary fiber features may influence fiber fermentability and the bacteria specificity”. It is necessary to explain how the degree of polymerization, particle size, solubility, viscosity, and other dietary fiber features may influence fiber fermentability and the bacteria specificity.

Response 1.1: Thanks for the good point. We now describe the potential mechanisms and cite the related references (Line 111-116).

Point 1.2: Moreover, the author referred that “dietary fiber can be classified into three types based on the physiological properties of their MU polymerization: 1) nonstarch polysaccharides (NSP) (MU≥10); 2) resistant starches (RS) (MU≥10), and 3) resistant/nondigestible oligosaccharides (ROS) (MU: 3-9) ”. When different types of dietary fiber are fermented by intestinal flora, what are the differences in intestinal flora that ferment the dietary fiber and the produced metabolites?

Response 1.2: We now make it clear that “subsequent investigations observed homogeneous fermentation and physiological activities of indigestible oligosaccharides and polysaccharides that contain similar monosaccharides” (Line 59-61). We also list the intestinal flora that ferment the three different types of dietary fiber in Figure 1, which were briefly described at Line 180-185.

Point 2: In the part of “5. Impacts of dietary fiber on gut microbial composition and diversity”, it is necessary to introduce the specific changes of gut microbial composition.

Response 2: We regarded the composition as the overall community structure of gut microbiota, usually measured by beta diversity. To avoid misunderstanding, we now use the words “community structure” instead of the word “composition”.

Point 3: In the part of “6. Influences of dietary fiber on different gut microbes”, the authors only introduced the changes of gut microbes. How about the changes of metabolites produced by gut microbes? What is the significance of these changes and what do these changes mean?

Response 3: Thanks for suggestion. Considering that SCFAs are the major metabolites produced by gut microbes fermenting fiber, and few studies focused on other metabolites, we just summarized the changes of the metabolite in the part of “4. Fermentation of dietary fiber by the gut microbiota”. We now move the statement from the part 4 to the part 6. (Line 233-236).

Point 4: In the part of “Table 2”, the result description in table 2 is too simple.

Response 4: We now present all cut-off thresholds of dietary fiber for altered taxa of gut microbiota in Table 2.

Point 5: In the part of “7. Cut-off threshold of dietary fiber intake on gut microbiota”, only the cut-off threshold of Bifidobacterium was introduced. How about other gut microbiota?

Response 5: The threshold of fiber for Bifidobacterium was described in most studies, while those for other taxa were not. As suggested, we now add the information to the manuscript (Line 288-290).

Point 6: In the part of “Modulation of dietary fiber on gut microbiota in diabetes patients”, Since alterations in the human gut microbiome have been implicated in a wide range of complex and chronic conditions, including obesity, diabetes, cancers, and cardiovascular disease, why did the authors only introduce the modulation of dietary fiber on gut microbiota in diabetes patients? How does dietary fiber modulate gut microbiota in patients with other diseases, such as obesity, cancers, and cardiovascular disease?

Response 6: In this review, we took type 2 diabetes as an example to show the fiber-induced modulation on gut microbiota in patients with chronic diseases. We now make it clear in the section of “8. Modulation of dietary fiber on gut microbiota in diabetes patients” (Line 342-346).

Reviewer 4 Report

I congratulate the authors for the manuscript, the clarity in the exposition and the representation of the tables, one important observation concerns the updating of the classification

species names should be changed according to the new classification of the former Lactobacillus genus (Zheng et al., 2020.International journal of systematic and evolutionary microbiology,70, 2782-2858

Author Response

Point 1: I congratulate the authors for the manuscript, the clarity in the exposition and the representation of the tables, one important observation concerns the updating of the classification species names should be changed according to the new classification of the former Lactobacillus genus (Zheng et al., 2020. International journal of systematic and evolutionary microbiology,70, 2782-2858)

Response 1: Thanks for the point. We double check all articles cited in this review and find one involving newly classified species of Lactobacillus genus. We now add the information at Line 315-316.

Round 2

Reviewer 3 Report

The authors have revised the manuscript according to the reviewer's comment, it can be accepted in present form.